# The Effect of Transcranial Direct Current Stimulation (tDCS) on Anorexia Nervosa: A Narrative Review

**DOI:** 10.3390/nu15204455

**Published:** 2023-10-20

**Authors:** James Chmiel, Anna Gladka, Jerzy Leszek

**Affiliations:** 1Institute of Neurofeedback and tDCS Poland, 70-393 Szczecin, Poland; 2Department and Clinic of Psychiatry, Wrocław Medical University, 54-235 Wrocław, Poland

**Keywords:** tDCS, transcranial direct current stimulation, anorexia nervosa, non-invasive brain stimulation, neurostimulation, neuromodulation, eating disorder

## Abstract

(1) Introduction: Anorexia nervosa (AN) is a severe, debilitating disease with high incidence and high mortality. The methods of treatment used so far are moderately effective. Evidence from neuroimaging studies helps to design modern methods of therapy. One of them is transcranial direct current stimulation (tDCS), a non-invasive brain neuromodulation technique. (2) Methods: The purpose of this narrative review is to bring together all studies investigating the use of tDCS in the treatment of AN and to evaluate its effect and efficiency. Searches were conducted in the Pubmed/Medline, Research Gate, and Cochrane databases. (3) Results: The literature search resulted in five articles. These studies provide preliminary evidence that tDCS has the potential to alter eating behaviour, body weight, and food intake. Additionally, tDCS reduced symptoms of depression. Throughout all trials, stimulation targeted the left dorsolateral prefrontal cortex (DLPFC). Although the number of studies included is limited, attempts were made to elucidate the potential mechanisms underlying tDCS action in individuals with AN. Recommendations for future tDCS research in AN were issued. (4) Conclusions: The included studies have shown that tDCS stimulation of the left DLPFC has a positive effect on AN clinical symptoms and may improve them, as measured by various assessment measures. It is important to conduct more in-depth research on the potential benefits of using tDCS for treating AN. This should entail well-designed studies incorporating advanced neuroimaging techniques, such as fMRI. The aim is to gain a better understanding of how tDCS works in AN.

## 1. Introduction

Anorexia nervosa (AN) is a common and severe mental illness with multifactorial etiopathogenesis [1]. It transcends geographical and socioeconomic boundaries, affecting individuals worldwide. AN patients exhibit an obsession with food, leading to drastic food intake restrictions and a refusal to maintain a healthy body weight, as evidenced by a BMI below 18.5 kg/m2 [2]. Common eating behaviours among individuals with AN include prolonged meal gaps, calorie counting, selective eating schedules, and avoidance of certain food types [3]. Predisposing factors for AN include features of excessive cognitive control, such as obsessive–compulsive personality disorder, perfectionism, and a distorted perceived image of body shape or weight [4]. Patients experience intense fear and preoccupation concerning weight gain and body image.

The prevalence of AN is associated with a high morbidity rate [5]. This eating disorder imposes a substantial burden on physical and psychosocial well-being, with prevalence rates reaching as high as 4% in adolescent girls and young adult women [6]. It is important to note that AN is less commonly observed in men [7,8], with a lifetime prevalence rate that might be up to 0.3% [9], though this does not diminish the burden on male patients [10]. In women, significant weight loss can lead to amenorrhea, which is characterised by the absence of menstrual periods [11]. The cause of this phenomenon is believed to be dysfunction of the hypothalamic–pituitary axis [12]. However, in 2013, the American Psychiatric Association revised the diagnostic criteria for AN. They eliminated the strict weight requirements and the necessity for amenorrhea as a condition for diagnosis. This change stemmed from the recognition that hormonal changes in AN can vary, and some women, despite having low weights and exhibiting all the psychological symptoms of the disorder, may still experience regular menstrual cycles [12].

The availability of effective AN treatment methods is limited [13]. Cognitive-behavioural, psychodynamic, and family therapies are considered the treatments of choice and show moderate effectiveness [11]. Pharmacological therapy also has a limited role [14,15]. Selective serotonin reuptake inhibitors (SSRIs) and neuroleptics represent the primary treatment options, but their effectiveness is limited.

The outcomes of treatment for AN and the long-term prognosis are generally unfavourable [16]. AN has one of the highest mortality rates among all mental illnesses [17], with rates reaching up to 5.9% when considering all causes of death [18]. Notably, only about half of the patients achieve full recovery, and approximately one-third experience partial remission [11,19].

The Diagnostic and Statistical Manual of Mental Disorders, Fifth Edition (DSM-5), distinguishes between two clinical subtypes of AN: restrictive (AN-r) and binge eating/purging (AN-bp) [11]. Individuals with AN-r restrict their food intake and increase their physical activity, whereas those with AN-bp, in addition to restricting food, regularly engage in episodes of binge eating and/or purging behaviours [20]. These subtypes are characterised by distinct behaviours and personality traits. People with AN-bp tend to exhibit a greater inclination towards seeking novelty, sensation seeking, and higher impulsivity. On the other hand, individuals with AN-r demonstrate a lower tendency to seek novelty and exhibit higher levels of perseverance [21]. The presence of comorbidities can also vary between the subtypes. Those with AN-bp are more likely to experience substance abuse disorders, affective disorders, depression [22,23], impulsive behaviour [24], and borderline personality disorder [25]. In contrast, avoidant personality disorder is more commonly observed in individuals with AN-r [25].

The neurobiological basis of AN involves several neurotransmitters, including dopamine, serotonin, and norepinephrine, which play crucial roles in the development and maintenance of the disorder.

Dopamine is involved in modulating rewards and affect, and its dysregulation has been linked to the development of obsessive or ritualistic behaviours, such as the food rituals often seen in individuals with AN [26]. People with AN may exhibit impaired reward functioning, presenting as abstemious, anhedonic, and temperate in various behaviours from childhood, even before the onset of AN symptoms [27]. Dopamine is central in processing rewards, including food [28], and research has revealed altered striatal dopamine function in individuals with AN, particularly in response to highly palatable foods [29], possibly explaining their aversion to food. Unlike those without AN who experience pleasure from food, individuals with AN find it aversive, which might partially explain their persistent pursuit of self-starvation. Dopaminergic dysfunction also affects reward processing in situations unrelated to food, leading to difficulties in identifying the positive or negative value of stimuli [28].

Serotonergic dysfunction has been identified in AN [30]. Serotonin, a neurotransmitter known for its role in mood [31] and appetite regulation [32], is implicated in the disorder as a potential biological marker for AN [30]. Studies have linked certain genetic factors, such as the S allele of the 5-HTTLPR polymorphism of the serotonin transporter gene, to eating disorders, particularly AN [33]. Caloric restriction has a significant impact on serotonin availability in the brain, as tryptophan, a precursor to serotonin, is absorbed through food intake [34]. A restricted diet limits tryptophan availability, leading to decreased serotonin synthesis and increased oversensitivity to serotonin in postsynaptic receptors [34]. This depletion of serotonin may explain the high levels of anxiety and dysphoria often reported by individuals with AN. The dysregulation of serotonin receptors, particularly 5-HT1A and 5-HT2A, contributes to the persistence of AN symptoms and may explain the limited effectiveness of SSRIs in treating the disorder [28].

Norepinephrine, a neurotransmitter involved in regulating sympathetic arousal and anxiety [35], is elevated in individuals with AN, leading to heightened anxiety, particularly related to food and weight [28]. Early stages of dieting counteract this anxiety by depleting norepinephrine precursors typically obtained through food intake. This reduction in anxiety through dieting reinforces the behaviour, leading to further weight loss and the entrenchment of AN symptoms. Aberrant activity in the noradrenergic system is associated with irregular patterns of activation in the insula, a brain region implicated in AN [28].

The neurobiological basis of AN can be detected with fMRI. In individuals with the AN-restricting subtype, several studies have reported impaired brain function and metabolism at rest [36,37,38,39,40,41,42,43]. This dysfunction, characterised by reduced regional cerebral blood flow and a reduced regional cerebral metabolic rate of glucose, is evident in three key brain regions: the cingulate cortex (including Brodmann areas 24 and 32) [37,38,39,41], the parietal cortex (including the inferior parietal lobule) [42,43], and the frontal cortex (including the dorsolateral prefrontal cortex) [36,37,38,39,42,43]. In the AN binge–purge subtype, studies have demonstrated functional and metabolic reductions at rest in the bilateral frontal cortex [36,39,44]. Interestingly, these reductions have been shown to normalise after recovery [40]. Additionally, bilateral reductions in parietal metabolism and function have been observed, which also return to normal following recovery [36,44,45]. Furthermore, two studies have reported bilateral metabolic and functional reductions in the cingulate cortex, which also normalise after recovery [39,41].

Due to the high mortality rate associated with AN and the moderate effectiveness of currently utilised treatment methods, there is an obligation for the development and implementation of novel therapeutic approaches. Considering that AN is typified by alterations in brain neurobiology, and given the fMRI evidence pointing to reduced activity in specific brain regions, the application of a non-invasive brain stimulation technique, such as transcranial direct current stimulation (tDCS), holds promise for potential utility and effectiveness.

tDCS is a brain stimulation tool that allows the stimulation of the cerebral cortex by means of two or more sponge electrodes with opposite polarities (anode and cathode), soaked in saline and applied to the scalp [46]. It is painless, well tolerated, and safe, with no or few side effects [47]. The stimulator is a battery-powered device that delivers a small amount of direct current (usually 0.5–2 mA), some of which reaches the brain. Brain modulation depends on the polarity of the applied current. tDCS allows two types of stimulation: anodal and cathodal [48,49]. During tDCS, a current flows between the electrodes, making its effects non-focal. However, modifying the electrode size can enhance its focus [50]. Stimulation durations typically range from 15 to 30 min, with 20 min being the most common protocol. Anodal tDCS depolarises neurons, making them more excitable and facilitating firing, whereas cathodal tDCS hyperpolarises neurons, inhibiting firing below the stimulation site. Usually, target neurons are less excitable, and their spontaneous activity decreases [51]. The neurobiological effects of tDCS can persist beyond the stimulation period when applied for at least three minutes [48]. If tDCS is administered for more than 10 min with a current of 1 to 2 mA, the changes remain stable for at least one hour [52]. A single session of tDCS, lasting up to 15 min, has an impact on cortical excitability for approximately 90 min. Repeated stimulation can further extend this effect. The prolonged influence of tDCS on cortical excitability is associated with synaptic modulation mechanisms, as indicated by studies conducted on humans [53] and animal models [54,55]. tDCS induces calcium-dependent glutamatergic synaptic plasticity. The secondary effects of tDCS, both anodal and cathodal, can be prevented by blocking the N-methyl-D-aspartate (NMDA) receptor, but they can be enhanced by appropriate receptor agonists [56,57]. Anodal and cathodal tDCS decreases GABAergic activity [58], which may serve as a mechanism to regulate tDCS-induced plasticity. tDCS affects the balance between cortical excitation and inhibition by modulating the levels of γ-aminobutyric acid (GABA), glutamate/glutamine, and BDNF [59]. Low calcium amplification of the postsynaptic neuron leads to long-term depression (LTD), whereas a high concentration results in long-term potentiation (LTP) [60].

tDCS is a non-invasive technique that has been extensively studied for its ability to modulate cortical activity in humans, affecting perceptual, cognitive, and behavioural functions. It has shown effectiveness in a wide range of neurological and psychiatric disorders. For example, neurodevelopmental disorders, including autism spectrum disorder (ASD), schizophrenia (SCZ), and attention deficit/hyperactivity disorder (ADHD), are all characterised by imbalances in excitatory (E) and inhibitory (I) neural activity. These imbalances can lead to cognitive deficits, behavioural problems, and other symptoms. GABA and glutamate are neurotransmitters that are crucial for regulating E/I balance. Changes in the levels and functioning of these neurotransmitters have been observed in individuals with neurodevelopmental disorders. tDCS can lead to long-term effects on neural plasticity and connectivity. It may modulate neurotransmitters like dopamine, acetylcholine, and serotonin and affect membrane ion channels. Changes in GABA and glutamate levels, as well as the balance between them, are associated with tDCS-induced plasticity. Studies suggest that tDCS has potential therapeutic benefits for neurodevelopmental disorders. It can help restore E/I balance, improve cognitive function, and alleviate symptoms like social deficits, hallucinations, inattention, and impulsivity [61]. To date, several reviews have examined the effectiveness of tDCS in the treatment of AN [62,63,64,65,66]. However, they covered all eating disorders, including bulimia, binge eating, and food craving. AN itself is a heterogeneous disorder, and even more so, individual eating disorders are heterogeneous. Conducting such a general overview of all eating disorders may bias the results.

This narrative review focuses specifically on the application of tDCS to alleviate AN symptoms. tDCS can influence neurofunctional reorganisation and behavioural changes by altering cortical excitability. Given that AN is also associated with underactivity of various brain areas, it is well suited for neuromodulatory treatment.

## 2. Methods

### 2.1. Data Sources and Search Strategy

For this narrative review, J.C. and A.G. performed an independent online search using predefined criteria. The search combined the keywords ‘transcranial direct current stimulation’ or ‘tDCS’ with ‘anorexia’ or ‘anorexia nervosa.’ We considered publications in the Pubmed/Medline and Research Gate databases, with an access date of May 2023 and publication dates ranging from January 2008 to December 2022.

### 2.2. Study Selection Criteria

The eligibility criteria included clinical trials conducted in English during the specified period, investigating the effects of tDCS on AN. The exclusion criteria encompassed articles that were not published in English, reviews, and studies that did not use standardised psychological questionnaires.

## 3. Results

The screening process is represented in a flow chart (Figure 1). Initially, 8537 records underwent screening, with 8518 being excluded based on the evaluation of their titles and abstracts, primarily due to their topic relevance. Through the search strategies carried out in the database, 19 studies were identified. Of these, 14 studies were excluded on the grounds of their publication type. Following a comprehensive analysis of study titles and abstracts, 5 articles were deemed eligible for inclusion.

The studies that were found were published between 2014 and 2021. A total of 83 patients were enrolled (active tDCS = 50, sham tDCS = 21). One study was a randomised controlled trial (RCT), one study was single-blind and controlled, 2 studies were open-label, and one was a case report. Random assignment occurred in one study, and this particular study used sham stimulation for the control group. However, it is worth noting that the RCT study by Baumann et al. and the study by Costanzo et al. did not assess the blinding procedure.

### 3.1. Summary of Included Studies

The included studies are summarised in Table 1. In the study conducted by Khedr et al. [67], a total of seven patients participated in an open-label, single-arm experiment. Evaluations of tDCS effectiveness were carried out at three different time points: prior to tDCS sessions, immediately following the session, and one month later. Six patients used SSRIs during the study. The following measures were used: the Eating Disorder Inventory (EDI) to measure eating behaviour, the Eating Attitude Test (EAT-40) to measure symptoms of AN, and the Beck Depression Inventory (BDI) to measure symptoms of depression.

Baumann et al. [68] conducted a double-blind, randomised controlled trial to examine the effects of tDCS on eating behaviour, body weight, and depression in inpatients with AN. A total of 43 participants with AN were randomly assigned to receive either active tDCS (n = 22) or sham tDCS (n = 21). Outcome measures included the Eating Disorder Examination Questionnaire (EDE-Q) to measure eating psychopathology, the Zung Self-Rating Depression Scale (ZUNG) to measure depression, and Body Mass Index (BMI) to measure body weight. These assessments were conducted at four stages: (1) before tDCS treatment, (2) after tDCS treatment, (3) two weeks after the treatment, and (4) four weeks after the treatment. During treatment, patients took medication, mainly antidepressants and antipsychotics.

The aim of the study conducted by Costanzo et al. [69] was to investigate the potential of tDCS in modifying or resetting inter-hemispheric balance in adolescents with AN to restore control over eating behaviours. Twenty-three adolescents with AN participated in the study and received treatment as usual (AU), which included nutritional, pharmacological, and psychoeducational treatment. Additionally, they underwent either 18 sessions of tDCS combined with AU (tDCS + AU group, n = 11) or family-based therapy combined with AU (FBT + AU group, n = 12). All participants underwent clinical examinations to assess their mental health conditions. The evaluation of AN-related symptoms included measures such as the Eating Disorder Inventory (EDI-3), Eating Attitudes Test (EAT-26), and Body Uneasiness Test (BUT). Anxiety and depressive symptoms in the children were examined using the Multidimensional Anxiety Scale for Children (MASC) and the Children’s Depression Inventory (CDI). BMI was also measured. Additionally, each participant received atypical antipsychotic medication, with aripiprazole being the specific drug used for treatment. Additionally, some individuals in the study were prescribed SSRIs (five in the tDCS group and nine in the FBT group), whereas a few received benzodiazepines (two in the tDCS group and one in the FBT group) as well.

Strumila et al. [70] conducted a pilot study to assess the effects of tDCS in a group of nine female patients with AN. During two weeks of stimulation, none of the participants underwent any specific re-feeding protocol or nutritional intervention, nor did they attend specialised psychological intervention groups. All the patients were prescribed a range of different types of psychiatric medication. The study measured the EDI, EDE-Q, Body Shape Questionnaire (BSQ-34), and BDI.

Rzad et al. [71] presented a case report involving the application of tDCS on an 18-year-old female participant. The study aimed to assess several aspects, including anthropometric measures, biological factors, and psychological aspects. Anthropometric measures were evaluated using a bioelectrical impedance analysis, and fasting venous blood samples were taken to analyze certain biological factors. The patient was taking sertraline (SSRI) during treatment. The study also featured a battery of psychological assessments, including the Eating Attitudes Test (EAT), the Rosenberg Self-Esteem Scale (RSS), the Beck Depression Inventory (BDI), the Eating Disorder Examination Questionnaire (EDE-Q), the Body Esteem Scale (BES), and the Perceived Stress Scale (PSS). Furthermore, Body Mass Index (BMI) was measured as part of the comprehensive evaluation.

### 3.2. Technical Aspects of tDCS in AN and Safety

The studies included in this review used different methodologies, which are discussed in detail based on the work of Thair et al. [72].

The most common electrode montage was bipolar, with both the anode and cathode electrodes directly placed on the brain and an equal current applied to both. In another study, a monopolar montage was employed, in which one electrode was positioned on the scalp and the other was positioned externally, for instance, on the arm.

All the studies included in this research employed anodal stimulation on the left dorsolateral prefrontal cortex (DLPFC). Four of the studies [68,69,70,71] followed the international 10–20 system to position the electrodes accurately. This method involves measuring the participant’s head to identify specific regions of interest, and based on this, the anodal electrode was placed at F3. In one study [67], the left DLPFC was located by measuring the head 6 cm anterior to the left primary motor cortex along a parasagittal line. Regarding the reference (cathodal) electrode, its placement differed among the studies. In one study [67], the cathodal electrode was applied on the contralateral arm (extracephalic). Other studies once again employed the 10–20 system. In one study [68], the electrode was placed over the right orbitofrontal region (Fp2), and in three other studies [69,70,71], it was positioned over the right DLPFC (F4). The studies used different current intensities. One study [69] used 1 mA, whereas the others [67,68,70,71] used 2 mA. The duration of stimulation also varied. In some studies [67,70,71], it lasted 25 min. In one study [68], it lasted 30 min, and in another study [69], it lasted 20 min. Additionally, the frequency of stimulation differed among the studies. In two studies [67,68], tDCS sessions were administered daily for ten consecutive days, with five sessions per week. One study [69] utilised 18 sessions, three times a week, over a total of six weeks. In the case of two studies [70,71], they used 20 sessions, with two sessions per day for two weeks. Different sizes of electrodes were employed across the studies. One study [67] used a smaller anode (24 cm^2^) and a larger cathode (100 cm^2^). Similarly, another study [68] used an anode of 25 cm^2^ and a cathode of 51 cm^2^. In contrast, one study [69] used electrodes of the same size (25 cm^2^). However, the size of the electrodes was not specified in two studies [70,71].

All five studies reported mild adverse effects associated with tDCS treatment. These effects included tingling, headaches, heaviness in the head, itching, and a scalp burning sensation. In the study by Khedr et al. [67], only transient local itching was reported by two patients, and no other adverse effects were observed. Costanzo et al. [69] found that the most common adverse effects were an itching sensation and burning sensation, reported by nine participants, especially during the initial seconds of stimulation. These sensations rapidly diminished with the addition of water with a sponge. Local redness was also reported by eight participants. Other effects, such as headaches and tingling, were reported by five participants.

In Strumila et al. [70] study, two-thirds of the participants experienced light redness and burning, which may be partially attributed to the skin fragility associated with AN.

Baumann et al. [68] reported various adverse effects, including a burning sensation (six patients), headache (four patients), tingling and itching (three patients), fatigue, acute mood changes, and pinching (two patients). One patient reported additional symptoms such as stitching, pressure in the head, blurred vision, scalp pain, hyperglycemia with the onset of diabetes mellitus type I, burning in the eyes, twitching of the eye, and a positive mood.

### 3.3. Effects on Psychopathology and Eating Behaviour

In the study by Khedr et al. [67], the first, third, and fourth patients showed improvement in the EDI. Among the subscales of the EDI, the most prominent changes were observed in body dissatisfaction, interpersonal distrust, interoceptive awareness, and ineffectiveness.

In the study by Baumann et al. [68], the primary analysis, based on ANOVA results, did not reveal any significant effects of tDCS on complex psychopathology and weight recovery in patients with AN as measured by the EDE-Q. However, the secondary analysis indicated a potential positive impact of tDCS treatment on questions 4 and 23 of the EDE-Q. In comparison to sham tDCS, active tDCS resulted in a significant improvement in self-evaluation based on body shape and a significant decrease in the need for excessive control over calorie intake during the four-week follow-up. It is important to note that these results did not survive multiple comparison correction.

In Costanzo et al.’s [69] study, both groups showed improvement across various EDI-3 subscales. Regardless of the group, a reduction in the mean scores was observed after treatment in the following areas: drive for thinness, body dissatisfaction, eating disorder risk, low self-esteem, personal alienation, interpersonal insecurity, interpersonal alienation, asceticism, ineffectiveness, interpersonal problems, and global psychological maladjustment (all *p* < 0.05). However, there was no noticeable improvement in the subscales of bulimia, interoceptive deficits, emotional dysregulation, perfectionism, fear of maturity, affective problems, and overcontrol (all *p* > 0.10). There was no improvement in BUT.

In the study by Strumila et al. [70], EDI scores decreased significantly, with a large effect size of 0.62. Additionally, the scores decreased significantly in the following subdimensions: inefficiency, perfectionism, distrust, interoceptive awareness, fear of maturity, and asceticism, with the mean effect size for all those items being around 0.45, indicating high efficacy. The remaining measures were not assessed immediately after the completion of the treatment.

In the study by Rzad et al. [71], improvement was observed in the RSS, EDE-Q, and BES scales.

### 3.4. Effects on AN Symptoms

In the study by Khedr et al. [67], the first, third, and fourth patients showed improvements in EAT.

In the study by Costanzo et al. [69], there was an improvement in EAT.

In the study by Rzad et al. [71], there was an improvement in EAT.

### 3.5. Effects on Depressive Symptoms

In the study by Khedr et al. [67], the first, third, and fourth patients improved in the BDI-II.

In the study by Baumann et al. [68], the researchers anticipated some improvement in the active group based on the ZUNG. However, upon the completion of the treatment (stage 2), the sham group exhibited better results in the total score and specific questions (5, 11, 12, 20) of the ZUNG (*p* < 0.01). When comparing the first and last stages, the sham group demonstrated a significant decrease for questions 10 and 16 (*p* < 0.01 and *p* < 0.05). In the study by Costanzo et al. [69], improvement in the CDI was demonstrated.

In the study by Strumila et al. [70], BDI depression scores were significantly reduced (*p* < 0.01, effect size 0.47).

In the study by Rzad et al. [71], depression scores for the BDI were reduced.

### 3.6. Effects on Anxiety

In the study by Costanzo et al. [69], regarding the MASC subscales, irrespective of the group, there was a decrease in the mean scores after treatment for physical symptoms and ADI (all *p* < 0.05), whereas there was no evident improvement in harm avoidance, social anxiety, and separation/panic (all *p* > 0.10).

In the study by Rzad et al. [71], stress scores (for simplicity, we placed them in the “anxiety” category) were lowered.

### 3.7. Effects on BMI

In the study by Baumann et al. [68], in both the sham and active groups, BMI values showed improvement, although not significantly. In Costanzo et al. [69] study, they found that BMI showed a significant improvement in the tDCS + AU group (*p* < 0.001), whereas in the FBT + AU group, there was no significant change in BMI after treatment when compared to the baseline (*p* = 0.2). The average percentage increase in BMI [(T1 − T0)/T0 × 100] was 13.3% (±9.4) in the tDCS + AU group, whereas in the FBT + AU group, it was only 4.2% (±5.7).

In the study by Rzad et al. [71], an improvement and an increase in BMI were achieved.

### 3.8. Effects on Other Measurements

In the study by Rzad et al. [71], following the two-week stimulation period, improvements were observed in anthropometric measurements and certain blood parameters, such as ferritin levels. Notably, no significant adverse changes in blood parameters were observed as a result of the intervention.

### 3.9. Durability of tDCS Effects

The studies had different follow-up periods. The studies by Khedr et al. [67], Costanzo et al. [69], and Strumila et al. [70] measured outcomes 1 month after intervention, the study by Baumann et al. [68] measured outcomes after 4 weeks, and the study by Rzad et al. [71] measured outcomes after 2 weeks.

In the study by Khedr et al. [67], the first, third, and fourth patients maintained improvements for all three scales one month after completing treatment.

In Costanzo et al.’s [69] study, when they followed up with the tDCS + AU group one month after the treatment had ended, they found that the positive effects for most of the psychological measures still remained (EAT-26, T0 vs. T2: Z = 2.37, *p* = 0.02; MASC, T0 vs. T2: Z = 1.96, *p* = 0.05; CDI, T0 vs. T2: Z = 2.19, *p* = 0.04). However, there was no significant change in the EDI-3 scores (T0 vs. T2: Z = 1.33, *p* = 0.18). Furthermore, there was no improvement in the BUT scores even after one month (T0 vs. T2: Z = 0.56, *p* = 0.57). The improvement was also maintained for BMI.

In the study by Strumila et al. [70], EDI scores were maintained one month after stimulation. There was also a notable decrease in the overall score of the EDE-Q, and this reduction had a substantial effect, with a magnitude of 0.42. Comparable outcomes were seen in the EDE-Q restraint and eating concern subcategories. Scores for the BSQ-34 and BDI also remained lower.

In the study by Rzad et al. [71], there was sustained improvement (and further improvement in scores) in EAT, RSS, BDI, EDE-Q, BES, PSS, and BMI two weeks after the completion of tDCS.

## 4. Discussion

Studies have consistently demonstrated that tDCS stimulation of the left DLPFC has a positive effect on AN clinical symptoms and may improve them, as measured with various assessment measures. In this section, we discuss the mechanisms of action of tDCS in AN and propose additional parameters that are worth investigating in future trials.

### 4.1. General Findings

There is an increased interest in the use of tDCS to modulate eating behaviour, leading to a diverse range of methodological approaches. Although these approaches are crucial for the initial exploration of tDCS effects on various populations and measures, it is important to establish a robust foundation of studies [64]. Most of the studies that were analysed are open-label studies or case studies, which means their results should be interpreted with caution. Nevertheless, these studies provide preliminary evidence that tDCS has the potential to alter eating behaviour, body weight, and food intake. Therefore, a recommendation can be made for further research and development of tDCS protocols that can be used in the treatment of AN.

All studies involved anodal stimulation of the left DLPFC, a region that is relevant to AN’s pathophysiology. Current intensities varied from 1 mA for adolescents to 2 mA for adults, reflecting age-appropriate dosages. A lower current dose is usually used in children, but the stimulation is still effective. Stimulation frequency also varied. Once-daily stimulation is the standard approach and is the simplest to implement from an organisational perspective. However, in Strumila et al.’s [70] study, we can observe that, after twice-daily simulation during the one-month follow-up period, the improvement was not only maintained but also increased.

### 4.2. Impact on AN Symptoms, Psychopathology, and Eating Behaviours

The results from the included studies indicate that tDCS stimulation of the left DLPFC holds promise for influencing various aspects of AN symptoms, psychopathology, and eating behaviours. In the study by Khedr et al. [67], improvements were observed in EDI scores, specifically in areas such as body dissatisfaction, interpersonal distrust, interoceptive awareness, and ineffectiveness. These improvements were sustained one month after treatment. Only one patient who did not receive SSRIs showed no improvement. Additionally, the study reported improvements in the EAT, suggesting potential benefits in AN symptomatology, but only in combination with the use of SSRIs. Baumann et al. [68] reported mixed findings, with no significant effects on complex psychopathology and weight recovery as measured by the EDE-Q. However, a secondary analysis indicated potential positive impacts on specific aspects of body shape evaluation and the need for excessive calorie intake control, although these results did not survive multiple comparison correction. This suggests that tDCS may have nuanced effects on different aspects of AN. Costanzo et al.’s [69] study showed improvement across various EDI-3 subscales, indicating positive changes in aspects like drive for thinness, body dissatisfaction, and interpersonal issues. Both the tDCS + AU group and the family-based therapy + AU group showed improvements, emphasising the potential of tDCS as an adjunctive treatment. However, not all subscales showed improvement, highlighting the need for further research to pinpoint the specific areas where tDCS is most effective. Strumila et al.’s [70] study demonstrated significant reductions in EDI scores, indicating improvements in various aspects of psychopathology such as inefficiency, perfectionism, distrust, and fear of maturity. These changes were maintained one month after treatment. Rzad et al.’s [71] case report showed improvements in various psychological assessments, including the EAT, RSS, EDE-Q, BES, and PSS. These improvements suggest that tDCS may have a positive impact on a wide range of psychological factors associated with AN.

Overall, although these studies provide encouraging preliminary results, it is important to acknowledge the limitations of small sample sizes, open-label designs, and variations in tDCS protocols across studies. Further randomised controlled trials with larger sample sizes and standardised protocols are needed to confirm the efficacy of tDCS in AN treatment and to better understand the specific mechanisms underlying its effects. Additionally, future research should explore the potential long-term effects of tDCS treatment beyond the one-month follow-up period, as well as investigate the durability of improvements in AN symptoms and psychopathology over extended time frames. Moreover, the combination of tDCS with other therapeutic approaches, such as cognitive behavioural therapy or nutritional interventions, should be explored to determine if synergistic effects can be achieved in the treatment of AN.

### 4.3. Impact on BMI

Three of the included studies used BMI as outcomes and showed mixed results. In the study by Baumann et al. [68], both the active and sham tDCS groups showed improvement in BMI, although the improvement was not statistically significant. This suggests that tDCS may have a limited impact on BMI in this specific population, and additional factors such as nutritional interventions and psychotherapy may be necessary for significant weight gain. In the study by Costanzo et al. [69], the tDCS + AU group showed a significant improvement in BMI, with a notable average percentage increase. In contrast, the FBT + AU group did not exhibit a significant change in BMI after treatment. This suggests that tDCS, when combined with treatment as usual (AU), may have a beneficial effect on increasing BMI in adolescents with AN. In the case report by Rzad et al. [71], an improvement and increase in BMI were observed after tDCS treatment. Although this is a single case report, it still provides valuable insight into the potential of tDCS to positively influence BMI in AN. These mixed findings regarding the impact of tDCS on BMI highlight the complexity of AN treatment and the multifaceted nature of the disorder. It is possible that tDCS may have a more significant impact when combined with other therapeutic approaches, such as nutritional support and psychotherapy. Therefore, future research should explore the specific factors that contribute to changes in BMI in AN patients undergoing tDCS treatment.

### 4.4. Impact on Depression

Four studies investigated the effects of tDCS on depression in patients with AN, revealing mixed but predominantly positive results. In the studies conducted by Khedr et al. [67] and Rzad et al. [71], improvements in depression symptoms were observed following tDCS treatment. Khedr et al. reported that the first, second, third, and fourth patients in their study demonstrated improvements in BDI-II scores. This suggests that tDCS may have a positive effect on reducing depressive symptoms in AN patients but in combination with SSRIs.

In contrast, the study by Baumann et al. [68] presented more nuanced results. Although the researchers initially anticipated some improvement in the active tDCS group based on the ZUNG, they found that, upon the completion of the treatment, the sham group exhibited better results in the total score and specific questions of the ZUNG. Possible explanations for these findings include the presence of higher levels of MDD and higher doses of antidepressants, particularly mirtazapine, in the sham group, which could have influenced the results. Another possible interpretation is that, in individuals with AN, difficulties in experiencing and regulating emotions may arise as a result of the primary eating disorder pathology and could potentially intensify with age. Therefore, if the patients’ core difficulties did not undergo sufficient changes, their moods may have remained unaffected.

Costanzo et al. [69] also demonstrated improvements in depressive symptoms using the CDI in their study of adolescents with AN who received tDCS treatment. This suggests that tDCS may have a positive impact on alleviating depressive symptoms in younger AN patients. Strumila et al. [70] reported significant reductions in BDI depression scores after tDCS treatment, indicating an improvement in depressive symptoms. This aligns with the findings from Khedr et al. [67] and Costanzo et al. [69].

### 4.5. Mechanisms of Action

In patients with AN, there is evidence of fronto-temporal hyperactivity in the right hemisphere (RH). The EEG study by Grunwald et al. [73] demonstrated hyperactivation in the RH of individuals with AN. Similarly, the positron emission tomography (PET) study conducted by Galusca et al. focused on serotonin activity around 5-HT1A receptors in people with AN, revealing an increased number of serotonergic junctions, particularly in the fronto-temporal regions of the RH [74,75].

Anodal tDCS, which has an excitatory effect, enhances the excitability of the left DLPFC and helps restore the interhemispheric balance by counteracting overactivity in the right DLPFC. Additionally, the effects of tDCS may extend to distant brain structures associated with the site of stimulation, which are also implicated in the underlying mechanisms of AN [67].

The pathogenesis of AN involves imbalances in serotonergic signaling in the ventral striatum, potentially related to the aversive aspects of the disorder [75,76]. Another theory proposes disruptions in reward pathways, as alterations in the dopamine system can affect reward circuitry, leading to anxiety and dysphoric moods [75,77]. Wagner et al. [78] discovered a dopamine imbalance in the ventral striatum of AN patients, characterised by decreased neuronal activity and hyperexcitability of the caudate nucleus.

In individuals with eating disorders, reward pathways are activated by disease-related stimuli but may not respond as strongly to typical rewarding stimuli. Excessive activation of the ventral striatum occurs in response to body-weight-related stimuli. Tanaka et al. [79] demonstrated that tDCS stimulation of the cortex can modulate dopamine release in the striatum. Cathodal tDCS, but not anodal tDCS, led to increased extracellular dopamine levels in the rat striatum for over 400 min. These findings suggest that tDCS may directly or indirectly impact the dopaminergic system in the basal ganglia, potentially affecting certain pathophysiological mechanisms of AN, including mesocortical dopaminergic pathways and increased food intake [75]. However, it remains unclear whether a current of such intensity can effectively reach deeper brain structures in humans.

In the study by Fonteneau et al. [80], bifrontal tDCS was applied to the human brain to investigate its effects on subcortical dopamine transmission during and immediately after stimulation. Right anodal and left cathodal tDCS resulted in a significant increase in extracellular dopamine transmission in the striatum, which is involved in the reward–motivation network. Mesolimbic dopaminergic projections in the striatum play a crucial role in guiding eating behaviour by modulating appetitive motivational processes. It has been suggested that disturbances in dopaminergic reward pathways contribute to the pathogenesis of AN [75,81,82].

The DLPFC plays a significant role in regulating emotions. By stimulating this region, the desire for dietary behaviours and calorie restrictions can be reduced. AN is characterised by excessive cognitive control, with the DLPFC being a key component of the cognitive control system. Even after recovery, individuals with a history of AN often exhibit heightened cognitive control over reward processing. Given these observations, inhibiting the left DLPFC has the potential to alleviate excessive cognitive control in AN [68].

tDCS has proven to be effective in reducing depressive symptoms in individuals with AN. This is not surprising, as tDCS has been commonly studied in depression treatment, and anodal stimulation of the left DLPFC has shown promise in depression therapy [83,84]. The underlying hypothesis behind using tDCS for depression is that it targets dysfunctions in various cortical and subcortical regions, including the prefrontal cortex [85], amygdala [86], and hippocampus [87]. Depressed individuals often have imbalances in brain activity, with increased activity in the right cortex and decreased activity in the left cortex. tDCS works by enhancing activity in the left DLPFC while reducing activity in the right cortex, aiming to address the neurophysiological imbalances associated with depression [88]. Moreover, tDCS may have an impact on dopamine secretion in the striatum. Some studies have indicated that applying tDCS to the left DLPFC increases dopamine secretion [89,90], which is significant because depression has been linked to dopaminergic dysfunction [91,92,93]. This positive effect on dopamine production could contribute to alleviating depression symptoms in AN. However, more neuroimaging studies are needed to confirm prefrontal dysfunction in AN patients and the specific effect of tDCS on this area of the brain. Nonetheless, the potential of tDCS as a non-invasive and promising treatment for depression in AN warrants further exploration and investigation.

### 4.6. Safety and Acceptance

tDCS is safe and acceptable for patients. The main risks are associated with the device (burn and skin irritation), interference with the psychiatric treatment, and adverse effects on cognitive performance. Regarding AN, there have been limited safety and acceptability studies [66].

### 4.7. Ethical Issues

The ethical considerations for tDCS arise from the application of the therapy to a population of very vulnerable and physically frail patients with severe and persistent AN. In this case, their ability to make healthy decisions is limited. Ethicists studying tDCS have also raised concerns that tDCS could be perceived as a form of “mind control” that increases patients’ dependence and helplessness and reduces their sense of authenticity [94,95]. The scant literature to date examining the views of patients treated with tDCS shows that they are capable of understanding the issues of benefits and risks to their authenticity [65].

### 4.8. Review Limitations and Risk of Bias

It should be emphasised that the research was analysed in the form of a narrative review. This raises some concerns about the lack of objectivity. Due to the small number of studies, it is not possible to conduct a meta-analysis. However, this is the first review of research focused solely on AN. Previous work, as mentioned earlier, has also included other eating disorders and positively assessed the impact of tDCS. Eating disorders are heterogeneous, so generalising the effects and effectiveness of treatments for all disorders raises significant concerns about biased results. Reviews should cover only one disorder. Another limitation of this review is the potential impact of psychotropic medication in the included studies, which may have introduced bias into the results. According to Normann et al. [96], SSRIs can enhance the response to tDCS. In healthy individuals, the chronic intake of SSRIs increased facilitative plasticity in the visual cortex, converting inhibitory plasticity to facilitation [96]. Citalopram, an SSRI, was found to enhance and prolong the facilitation induced by anodal tDCS while converting cathodal tDCS-induced inhibition into facilitation [97]. Therefore, patients in the study by Khedr et al. [67] who were taking SSRIs might have had an enhanced response to tDCS and a more pronounced treatment effect. Conversely, drugs that interfere with dopaminergic signaling, such as antipsychotics, are believed to have a negative impact on tDCS plasticity [69,98]. Aripiprazole, the medication used by all participants in Costanzo et al.’s [69] study, is a partial agonist of the D2 receptor and differs from other antipsychotics that act as pure antagonists [99]. Aripiprazole may specifically enhance the hypothesised effect of tDCS in regulating the tonic dopamine (DA) component in the striatum, as it inhibits the phasic component while preserving the release of the tonic DA component to some extent [69,100].

### 4.9. Future Directions

The evidence to date suggests that, although tDCS has potential for the treatment of people with AN, much of this potential is yet to be discovered. There is still significant heterogeneity in response to treatment. The optimisation of protocols, patient selection, treatment goals, and intervention parameters remains to be explored. Optimising the protocols will require extensive research to address issues such as frequency, duration, intensity of stimulation, and target selection. Given that AN is a heterogeneous disorder, finding a single “optimal” protocol is unlikely. Using personalised interventions to target specific subgroups of patients will be a valuable option. It is crucial to gain a deeper understanding of neural, neurocognitive, and biological correlations, markers, and outcome predictors, as they can help provide personalised treatments and individualised protocols. Work is underway to develop a rationale for the use of neuromodulatory therapies based on evolving neural models of eating disorders. These advances and increased knowledge about neural networks and their interconnections may lead to the emergence of new hypotheses regarding the etiology and treatment of AN [65]. Trials involving multimodal neuroimaging, neurocognitive tasks, and clinical measures should be conducted [64]. Future trials should include larger samples to be sufficiently powerful.

Individuals with AN frequently suffer from comorbidities, with depression being the most common. As noted in a review [62], the mood component in the research raises a methodological problem: the simultaneous improvement of body weight and mood in people with AN makes it impossible to identify a specific effect on AN. The measurement of depressive symptoms with a subgroup analysis of non-depressed patients could help solve this problem [62].

The review [62] highlights an important factor that should be taken into account, which is the variability of brain states associated with the metabolic status. In individuals with AN, the nutritional status can exert a more pronounced impact on brain functioning compared to other mental disorders. The nutritional status has been found to influence treatment responses, particularly in relation to antidepressants, which are less effective in underweight patients [101]. Consequently, nutritional status may play a role in determining the response or non-response to treatment and can potentially confound the effectiveness of neuromodulation techniques.

tDCS technology continues to evolve and increasingly enables more precise targeting of treatment, probing deeper areas of the brain, using shorter and more powerful protocols, and stimulating multiple brain regions simultaneously. According to new evidence, these types of interventions may work synergistically when used with various forms of cognitive training, but this has not yet been explored in AN [102].

## 5. Conclusions

In conclusion, the small evidence base so far suggests that tDCS has a positive effect on AN symptoms and is an effective and safe tool in the treatment of AN, and its tolerance is high. However, there are many knowledge gaps regarding the optimisation of protocols and mechanisms of action. Additionally, the use of tDCS in conjunction with another intervention or alone raises questions. As advances in knowledge better elucidate the brain mechanisms of AN, there will be an opportunity to make better use of tDCS. To establish a stronger evidence base and validate the therapeutic efficacy in this disorder, further randomised controlled trials (RCTs) are imperative.

## Figures and Tables

**Figure 1 nutrients-15-04455-f001:**
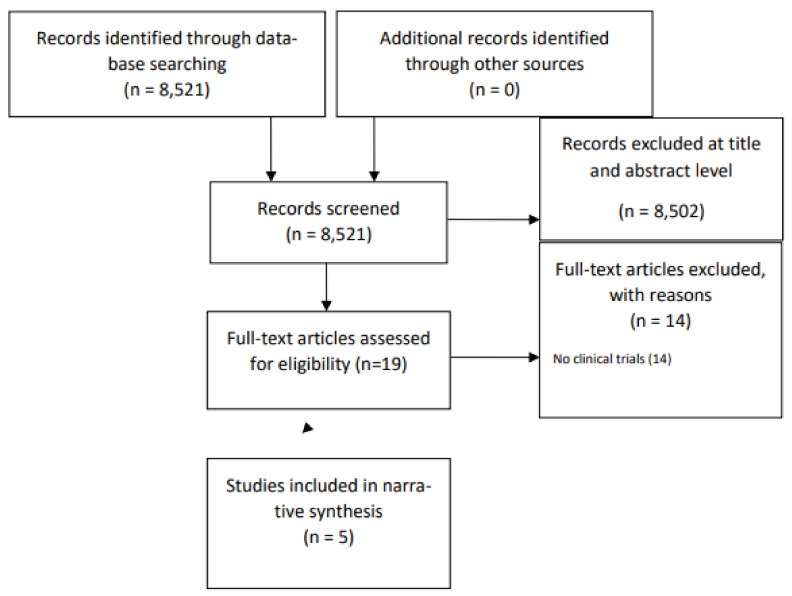
Flow chart depicting the different phases of the systematic review.

**Table 1 nutrients-15-04455-t001:** Summary of main findings from articles included in the review.

Author, Citation	Population	Technical Specifications	Outcome Measurment	Main Results (Primary Outcomes Are Bolded)
Khedr et al. [67]	7 patients with AN (n = 7).	Anodal tDCS (2 mA) over the left dorsolateral prefrontal cortex (DLPFC) for 10 consecutive days (5 session/week), anodal tDCS for 25 min.	Eating Attitude Test (EAT), Eating Disorder Inventory (EDI) and Beck Depression Inventory (BDI). Pre-tDCS, post-tDCS, and one month later.	3 patients improved in all three rating scales post tDCS and after 1 month; 1 patient improved only in the BDI; 2 patients showed improvement at the end of session but returned to the baseline after one month. The 7th patient had no changes.
Baumann et al. [68]	43 inpatients with AN, active (n = 22) or sham (n = 21) tDCS.	2 mA anodal stimulation over the left DLPFC with the cathode over the right orbitofrontal region, 10–30 min sessions.	Eating Disorder Examination Questionnaire (EDE-Q), Zung Self-Rating Depression Scale (ZUNG), BMI; pre-tDCS, post-tDCS, and one month later.	No significant effect on complex psychopathology and weight recovery in patients with AN.
Costanzo et al. [69]	23 adolescents with AN, tDCS + therapy as usual (tDCS + (AU) n = 11) or a family-based therapy (FBT + AU n = 12).	1 mA, anodal electrode positioned over the left DLPFC and cathodal electrode over the right DLPFC; 20 min, 3 times a week for 6 weeks.	EDI-3, Bulimia (B), Global Psychological Maladjustment (GPM), Interpersonal Problems, BMI. Pre-, post-, one month, and 6 weeks after tDCS.	After 4 and 6 weeks, BMI increased in the tDCS group; in this group, a medium negative correlation was found between improvements in BMI, B and GPM.
Strumila et al. [70]	9 female patients with AN (n = 9).	Anodal 2 mA stimulation, anode on the left DLPFC, and cathode on the right DLPFC, 2 times per day for 25 min, 2 weeks.	Eating Disorder Inventory (EDI), Eating Disorder Examination Questionnaire (EDE-Q), Body Shape Questionnaire (BSQ-34), Beck Depression Inventory (BDI). Pre-, post-, and one month after tDCS.	Depression symptoms significantly decreased post and after 1 month. EDI decreased significantly post and 1 month after stimulation, EDE-Q questionnaire at 1 month was significantly lower.
Rząd et al. [71]	1 patient with AN (n = 1).	Anodal 2 mA stimulation, anode on the left DLPFC, and cathode on the right DLPFC, twice daily for 25 min for 2 weeks.	Fasting venous blood; Eating Attitudes Test, Rosenberg Self-Esteem scale, BDI, EDE-Q, Body Esteem Scale, Perceived Stress Scale. Pre-, post-, and 2 weeks after tDCS.	Improvement in anthropometric measurement, some blood parameters (e.g., ferritin), symptoms of depression and stress, and self-body image after two weeks

## Data Availability

No new data were created or analysed in this study. Data sharing is not applicable to this article.

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
