# Peer review of "The Effect of Transcranial Direct Current Stimulation (tDCS) on Anorexia Nervosa: A Narrative Review"

_nutrients, 2023, doi:10.3390/nu15204455_

Round 1

Reviewer 1 Report

Dear Author, I appreciated the hard work behind you manuscript and I consider the topic of clinical interest. Nevertheless, I find major flaws that, in my opinion, compromise the manuscript publication. 

First, the manuscript would need an extensive and thorough english revision.

Also, the logic of the argumentation should be revised and rearrenged, in order to gain readability. 

In the introduction section, the description of AN phenomenology is partial and reductive. For example, the Authors cite a high morbidity rate, without further details. Also, the burden on female patients is mentioned, stating that the prevalence of AN is lower for males. This does not decrease the burden of male patients. 

Also, amenorrhea is cited as relevant: it can be a side effect of denutrition, but it has been excluded by the diagnostic criteria of AN. This sould be clarified.

The paragraphs were the neural basis of AN are presented are confusing: data and results should be re-organized. 

Regarding methodlogy, even if studies included were few, at least a systematic review, without meta-analysis, would have been desirable. The results section is a mere list of the included studies summaries. 

The aim of a review should be to provide clinical indications on treatments, and even a systematic review without meta-analisys can do that. In this paper, I cannot find any conclusive indication on the use of tDCS in AN patients. The discussion adds novel information and studies, not pertinent with the results of the review. 

Depression was given a particular importance; however, since the Authors declare that one strength of their work is to have focused on AN only, and not on eating disorders in general, this should also apply for depression. I would suggest to eliminate "depression" from the title and to argue on depression in the manuscript as related to AN only. 

As I stated before, it is my opinion that the manuscript would benefit from an extensive thorough english revision, and from a thorough manuscript editing. 

Author Response

Dear Author, I appreciated the hard work behind you manuscript and I consider the topic of clinical interest. Nevertheless, I find major flaws that, in my opinion, compromise the manuscript publication. 

Dear Sir or Madam Editor,

Thank you very much for your thorough analysis of my text. Your comments contributed to improving the manuscript, thanks to which we will be able to publish a valuable article that - hopefully - will contribute to the popularization of tDCS in the treatment of AN. The work was significantly rebuilt because its readability was very low - the text was chaotic, there was no separation between the results and the discussion, both sections mixed with each other. I fixed this error and hopefully the text is now better and more readable. I have responded to all comments below.

First, the manuscript would need an extensive and thorough english revision.

Ad. 1. The work has been checked by a native speaker.

Also, the logic of the argumentation should be revised and rearrenged, in order to gain readability. 

Ad. 2 The logic of the argument was changed, e.g. the 8th paragraph in the introduction was rebuilt.

In the introduction section, the description of AN phenomenology is partial and reductive. For example, the Authors cite a high morbidity rate, without further details. Also, the burden on female patients is mentioned, stating that the prevalence of AN is lower for males. This does not decrease the burden of male patients. Also, amenorrhea is cited as relevant: it can be a side effect of denutrition, but it has been excluded by the diagnostic criteria of AN. This sould be clarified.

Ad 3. Dear Editor, the work describes the details of the high morbidity rate - that it is 4% in teenage girls and young women. However, we added an incidence rate in men. We changed the meaning of the sentence in the second paragraph - previously it was said that AN is a burden for teenage girls and young women, which is illogical, because AN is a burden for all patients. We added information that the lower occurrence of AN in men does not reduce the burden of the disease. We added the probable cause of amenorrhea and information that this criterion was excluded from the diagnosis, and we also provided the reason.

The paragraphs were the neural basis of AN are presented are confusing: data and results should be re-organized. 

Ad. 4 Two paragraphs on the neural and neurobiological basis of AN have been expanded and reordered. Duplicate information removed in paragraph 7.

Regarding methodlogy, even if studies included were few, at least a systematic review, without meta-analysis, would have been desirable. The results section is a mere list of the included studies summaries. 

Ad. 5. We decided to conduct a narrative review due to the small number of studies and the possibility of a freer discussion. We have changed the “Methods” section to be consistent with the methodology for writing systematic reviews. We wrote the review in accordance with SANRA guidelines.

The aim of a review should be to provide clinical indications on treatments, and even a systematic review without meta-analisys can do that. In this paper, I cannot find any conclusive indication on the use of tDCS in AN patients. The discussion adds novel information and studies, not pertinent with the results of the review. 

Ad. 6 We have removed the previous sections 4.3. and 4.4. We wrote it because we are enthusiasts of tDCS and we believe that these two potential cognitive aspects should be explored in future studies, as improvements in these cognitive domains have been shown in healthy people. Now, in the discussion, a lot of attention was devoted to the analysis of the results and their significance. Interpretation removed from the results section.

Depression was given a particular importance; however, since the Authors declare that one strength of their work is to have focused on AN only, and not on eating disorders in general, this should also apply for depression. I would suggest to eliminate "depression" from the title and to argue on depression in the manuscript as related to AN only. 

Ad. 7. We have changed the title, in the text depression is one of many areas of action of tDCS in AN, not the main one.

Reviewer 2 Report

The article is a review of the literature on the use of transcranial direct current stimulation (tDCS) for treating anorexia nervosa (AN) and its comorbid depression. It discusses how tDCS, a non-invasive brain stimulation technique, can modulate cortical excitability by applying a weak electric current to the scalp. AN, a severe mental illness, is characterized by restricted food intake and distorted body image. The neurobiological basis of AN involves altered brain function in regions like the frontal, parietal, and cingulate cortex, along with dysregulation of neurotransmitter pathways. The rationale for using tDCS in AN is based on its potential to impact craving regulation, cognitive control, reward processing, and self-regulation. Reviewed studies suggest that tDCS may alter eating behavior, body weight, food intake, and reduce depression symptoms, primarily targeting the dorsolateral prefrontal cortex. However, these studies have methodological limitations and parameter heterogeneity. Technical aspects and safety of tDCS are discussed, noting common parameters and mild, transient adverse effects. Ethical concerns related to tDCS application in a vulnerable population are considered, as well as its potential impact on patient authenticity and autonomy. The article concludes by suggesting future research directions, including protocol optimization, patient selection, treatment goals, intervention parameters, neural correlates, outcome predictors, and combination with other therapies. The summaries are presented very well and are bound together by relevant background and references. However, a few changes will serve the article well in terms of acceptance among a broader audience.

1. Please revise lines 18 and 19 in the abstract to improve legibility.

2. Consider including a schematic diagram of tDCS for the benefit of readers unfamiliar with the technique.

3. While the neurophysiological effects of tFCS are well-summarized, briefly introduce the mechanism of action of tDCS for various diseases to provide better clarity and context.

4. Enhance the conclusion of the study by Khedr et al. by providing a slightly more detailed summary.

5. In Table 1, specify the current used in the study conducted by Rzad et al.

6. In section 3-2, mention the medication(s) taken by the subjects. While this aspect has been discussed in later sections, since it affects outcomes it deserves mention here.

7. The initial statement in the discussion section does not align with the conclusions of the mentioned studies (lines 342-344). Consider revising these lines to present a more critical conclusion. Please make sure it is consistent with lines 350-352.

8. Please provide a separate reference for lines 388-390, apart from the Khedr reference.

9. After introducing an acronym for a term, refrain from spelling out the term. For example, in line 497, use "AN" instead of spelling out the term.

10. Highlight the strength of the evidence presented in the studies to encourage future research aimed at optimizing the use of tDCS in AN. While this can be aligned with the section discussing future research directions, it would be best placed just before the conclusion section to emphasize its importance.

Author Response

The article is a review of the literature on the use of transcranial direct current stimulation (tDCS) for treating anorexia nervosa (AN) and its comorbid depression. It discusses how tDCS, a non-invasive brain stimulation technique, can modulate cortical excitability by applying a weak electric current to the scalp. AN, a severe mental illness, is characterized by restricted food intake and distorted body image. The neurobiological basis of AN involves altered brain function in regions like the frontal, parietal, and cingulate cortex, along with dysregulation of neurotransmitter pathways. The rationale for using tDCS in AN is based on its potential to impact craving regulation, cognitive control, reward processing, and self-regulation. Reviewed studies suggest that tDCS may alter eating behavior, body weight, food intake, and reduce depression symptoms, primarily targeting the dorsolateral prefrontal cortex. However, these studies have methodological limitations and parameter heterogeneity. Technical aspects and safety of tDCS are discussed, noting common parameters and mild, transient adverse effects. Ethical concerns related to tDCS application in a vulnerable population are considered, as well as its potential impact on patient authenticity and autonomy. The article concludes by suggesting future research directions, including protocol optimization, patient selection, treatment goals, intervention parameters, neural correlates, outcome predictors, and combination with other therapies. The summaries are presented very well and are bound together by relevant background and references. However, a few changes will serve the article well in terms of acceptance among a broader audience.

Dear Sir or Madam Editor,

Thank you very much for your thorough analysis of my text. Your comments contributed to improving the manuscript, thanks to which we will be able to publish a valuable article that - hopefully - will contribute to the popularization of tDCS in the treatment of AN. The work was significantly rebuilt because its readability was very low - the text was chaotic, there was no separation between the results and the discussion, both sections mixed with each other. I fixed this error and hopefully the text is now better and more readable. I have responded to all comments below.

  1. Please revise lines 18 and 19 in the abstract to improve legibility.1.Done.
  2. Consider including a schematic diagram of tDCS for the benefit of readers unfamiliar with the technique.2. We don't want to include an image from tDCS in the text because we are concerned about copyright. There are thousands of tDCS graphics on the Internet.
  3. While the neurophysiological effects of tFCS are well-summarized, briefly introduce the mechanism of action of tDCS for various diseases to provide better clarity and context.3. Done, lines 184-197.
  4. Enhance the conclusion of the study by Khedr et al. by providing a slightly more detailed summary.4. Study by Khedr et al. is discussed briefly in Section 3.1, the exact results are in the following results subsections. It has been shown that improved results were achieved when patients used SSRIs.
  5. In Table 1, specify the current used in the study conducted by Rzad et al.5. Done. Additionally, we reported the current used by Costanzo et al.
  6. In section 3-2, mention the medication(s) taken by the subjects. While this aspect has been discussed in later sections, since it affects outcomes it deserves mention here.6. Dear Editor, drug information is now included in a brief overview of each study. Information about medications did not fit into the subsection on safety and technical aspects, hence this decision 
  7. The initial statement in the discussion section does not align with the conclusions of the mentioned studies (lines 342-344). Consider revising these lines to present a more critical conclusion. Please make sure it is consistent with lines 350-352.7. Dear Editor, we have decided to create a new discussion. The previous version of the work was chaotic - the results section contains the results and their interpretations, and the interpretation is in the discussion. Now the discussion section provides interpretations of the results.
  8. Please provide a separate reference for lines 388-390, apart from the Khedr reference.8. We have provided a new reference -Tuplin et al.
  9. After introducing an acronym for a term, refrain from spelling out the term. For example, in line 497, use "AN" instead of spelling out the term.9. Fixed, AN is everywhere now.
  10. Highlight the strength of the evidence presented in the studies to encourage future research aimed at optimizing the use of tDCS in AN. While this can be aligned with the section discussing future research directions, it would be best placed just before the conclusion section to emphasize its importance.10.Done. The discussion describes what the results mean, what they might mean, and we strongly suggest future research to understand mixed results or correlations.

Reviewer 3 Report

Understudied area. Good approach. Good searching and selecting methods. Writing is good. The Discussions are nicely done. This could be of interest for the readers 

Author Response

Dear Sir or Madam Editor,

Thank you very much for your thorough analysis of my text.

The other reviewers made many comments about the work, so I had to radically revamp this manuscript. I hope that these changes will contribute to better readability and reception, and ultimately - to the popularization of tDCS in the treatment of AN. With respect.

Round 2

Reviewer 1 Report

Dear Authors,

I sincerely thank you for the punctual response to my comments, and appreciate the revion of the manuscript you have made. 

I still find some minor issues with the english language and phrasing that should be addressed, maybe not only by an english native speaker, but also from an expert in scientific reporting. 

Thank you.

I still find some minor issues with the english language (for example, "[...] an urgent need [...]") and phrasing (e.g. in the new paragraph regarding the changes in the DSM-5 criteria) that should be addressed , maybe not only by an english native speaker, but also from an expert in scientific reporting. 

Author Response

Dear Sir or Madam Editor,

Thank you for your suggestions regarding the article. The work was thoroughly checked by an English native speaker and many corrections were made. I hope it's good now.

Best regards.